# Investigating the relationship between diversity and generalization in over-parameterized deep neural networks

Ruan P. Van der Spoel*[1,2,3] and Randle Rabe[1,2,3]

[1]Faculty of Engineering, North-West University, South Africa
[2]MUST Deep Learning
[3]Centre for Artificial Intelligence Research (CAIR), South Africa
{ruanvdspoel, rrabemust}@gmail.com

## Abstract

In ensembles, improved generalization is frequently attributed to *diversity* among members of the ensemble. By viewing a single neural network as an *implicit ensemble*, we perform an exploratory investigation that applies well-known ensemble diversity measures to a neural network in order to study the relationship between diversity and generalization in the over-parameterized regime. Our results show that i) deeper layers of the network generally have higher levels of diversity—particularly for MLPs—and ii) layer-wise accuracy positively correlates with diversity. Additionally, we study the effects of well-known regularizers such as Dropout, DropConnect and batch size, on diversity and generalization. We generally find that increasing the strength of the regularizer increases the diversity in the neural network and this increase in diversity is positively correlated with model accuracy. We show that these results hold for several benchmark datasets (such as Fashion-MNIST and CIFAR-10) and architectures (MLPs and CNNs). Our findings suggest new avenues of research into the generalization ability of deep neural networks.

## 1 Introduction

A complete understanding of why deep neural networks (DNNs) generalize well on unseen data remains an open problem in machine learning. For example, it is well-known that neural networks achieve good generalization despite interpolating their training data [1–3]. Yet, despite our limited understanding of generalization in DNNs, rapid progress has been made in developing methods, called *regularization*, that encourages generalization. Examples of these methods include *Dropout* [4], *weight decay* [5], *input or weight noise* [5].

A widely-used regularization method is *ensembling*, where the output of several models are aggregated to produce a final output. Crucial to the generalization ability of the ensemble is the *diversity* of the models [6–9]. Depending on the task,

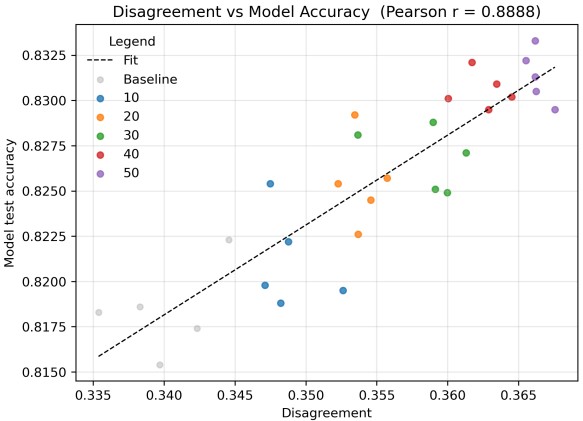

**Figure 1.** Model performance against average diversity (across layers) for models trained on the Fashion-MNIST dataset using DropConnect as a regularizer with values in the range $0-50$. Diversity is shown by the *Disagreement* measure. A higher *Disagreement* value constitutes a higher diversity. There is a clear increase in model performance as the average diversity increases.

there are several definitions of diversity that may be used [10]. For example, diversity can measure the correlation between the models' predictions in the ensemble, in which case, low correlation would be considered high diversity. In general, a higher diversity of the models will correspond to better generalization of the ensemble, albeit that there may be a trade-off where too much diversity can negatively impact generalization [6, 7, 11, 12].

Recently, an insightful approach to investigating generalization in deep learning models has been to view a single deep learning model as an *implicit ensemble*. For the problem of vanishing gradients in deep residual networks, Veit et al. [13] view a deep residual network as a collection of paths and show that the paths have an ensemble-like behavior. Another approach by Olson et al. [14] decomposes a single neural network into an ensemble of low-bias subnetworks and, by using correlation as a proxy for diversity and showing that the subnetworks exhibit low correlation, argues that an internal regularization process helps mitigate overfitting in neural networks. Through the observation of regularities in the activation patterns of the hidden nodes of a

---

*Corresponding Author.

Proceedings of the 7th Northern Lights Deep Learning Conference (NLDL), PMLR 307, 2026.

DNN, Davel et al. [15] views a single hidden node as a classifier. More recently, through investigating the problem of catastrophic forgetting in continual learning, Benjamin et al. [16] show that a neural network in the lazy regime can be decomposed into an implicit ensemble consisting of the weights of the neural network.

However, in viewing a neural network as an implicit ensemble, the role of *diversity* in a neural network remains unexplored. In this paper, we follow the approach of Davel et al. [15] and view a single neural network as an implicit ensemble with hidden nodes as classifiers. This allows us to investigate the role of diversity with respect to generalization of the network. The main contributions of the paper are as follows:

1. We examine the diversity of the hidden nodes using established diversity measures.
2. We empirically investigate the relationship between node-level diversity and generalization across several benchmark datasets and for different architectures.
3. We analyze the effect of well-known regularization methods that encourages generalization and examine the effect of these methods on diversity. We show that diversity correlates with the generalization of the neural network (see Figures 1 and 6).

Our results provide new insight into the ability of over-parameterized neural networks to generalize and offers new avenues of research into the generalization of neural networks.

# 2 Background

## 2.1 Ensemble Methods

Ensemble methods combine predictions of multiple classifiers to achieve better generalization than individual models. Classic approaches such as bagging [17] and boosting [18], have shown that ensembles reduce variance, improve robustness, and often outperform single models across diverse tasks.

A key factor to the success of ensembles is *diversity* among the base classifiers. If all classifiers make identical errors, the ensemble offers no advantage. However, when classifiers make different errors, the ensemble can correct individual mistakes, yielding improved accuracy [6, 7]. Theoretical and empirical studies have shown that ensembles benefit most when base learners are both accurate and diverse [10]. Diversity can be defined in several ways, for example it can be the degree of correlation between classifiers' predictions, with lower correlation implying higher diversity.

## 2.2 Diversity Measures

Several measures have been proposed to quantify diversity in ensembles [10]. In this paper, we focus on four well-known metrics: *Disagreement*, *Double-fault*, *Q-statistic*, and *Entropy*.

These measures (besides *Entropy*) are based on the outcomes of pairs of classifiers, which can be summarized using a $2 \times 2$ contingency table (Table 1). Assuming a $C$ class classification problem, let $D_a$ and $D_b$ be two classifiers. Given a data set $\{(x^{(i)}, y^{(i)})\}_{i=1}^{N}$, let $N^{11}, N^{10}, N^{01}$, and $N^{00}$ denote the frequency of the following cases:

- $N^{11}$: Number of samples for which both $D_a$ and $D_b$ are correct.
- $N^{10}$: Number of samples for which $D_a$ is correct, and $D_b$ is incorrect.
- $N^{01}$: Number of samples for which $D_a$ is incorrect, and $D_b$ is correct.
- $N^{00}$: Number of samples for which both $D_a$ and $D_b$ are incorrect.

Since every sample must fall into exactly one of these four categories, the sum of these four frequencies must equal the total number of samples ($N$) in the dataset:

$$N = N^{11} + N^{10} + N^{01} + N^{00}$$

We consider three pairwise metrics, namely:

- **Disagreement**

$$Dis_{a,b} = \frac{N^{10} + N^{01}}{N} \qquad (1)$$

We define the disagreement between two classifiers $D_a$ and $D_b$ based on the state of their prediction correctness on a sample $x^{(i)}$. This measures the proportion of instances where the classifiers exhibit a difference in their correctness status ($N^{10}$ or $N^{01}$). *Disagreement* is bounded between $[0, 1]$ where higher *Disagreement* values relate to higher diversity. A $Dis = 1$ indicates the highest amount of disagreement between classifiers.

- **Double-fault**

$$df_{a,b} = \frac{N^{00}}{N}, \qquad (2)$$

This measures the proportion of instances where both classifiers misclassify the same sample. *Double-fault* is bounded between $[0, 1]$, where a lower *Double-fault* value relates to a higher diversity (they fail on different samples).

- **Q-statistic**

$$Q_{a,b} = \frac{(N^{11}N^{00} - N^{10}N^{01})}{(N^{11}N^{00} + N^{10}N^{01})} \qquad (3)$$

Values range from $[-1, 1]$ with lower *Q-statistic* values indicating higher diversity.

**Table 1.** $2 \times 2$ contingency table for pairwise measures. See text for details.

|  | $D_b$ correct (1) | $D_b$ incorrect (0) |
|---|---|---|
| $D_a$ correct (1) | $N^{11}$ | $N^{10}$ |
| $D_a$ incorrect (0) | $N^{01}$ | $N^{00}$ |

We also consider one non-pairwise metric:

- **Entropy** *Entropy* quantifies the uncertainty in ensemble predictions and serves as a proxy for classifier disagreement [10, 19]. Let $P(c \mid x^{(i)})$ denote the proportion of classifiers in the ensemble that assign $x^{(i)}$ to class $c$, where $c \in \{1, \ldots, C\}$. This can be estimated as:

$$P(c \mid x^{(i)}) = \frac{1}{L} \sum_{m=1}^{L} \mathbf{1}\{D_m(x^{(i)}) = c\}, \quad (4)$$

where $L$ is the number of subpredictors, $D_m(x^{(i)})$ is the class predicted by the $m$-th classifier, and $\mathbf{1}\{\cdot\}$ is an indicator function which equals 1 if the condition inside is true, and 0 otherwise.. The per sample entropy is then calculated as:

$$H(x^{(i)}) = - \sum_{c=1}^{C} P(c \mid x^{(i)}) \log P(c \mid x^{(i)}). \quad (5)$$

We can then calculate the average over the dataset:

$$\text{Ent}_{\text{CC}} = \frac{1}{N} \sum_{i=1}^{N} H(x^{(i)}) \quad (6)$$

Higher values of *Entropy* indicate higher diversity between subpredictor predictions. It is important to note that *Entropy* uses the raw multi-class predictions, and not the binary correct/incorrect predictions (like the other measures do). This leads to Entropy being bounded between $[0, \log(C)]$ or $[0, 3.32]$ (the classification datasets we employ each consist of 10 classes).

These measures provide complementary views of diversity—*Disagreement* emphasizes complementarity, *Double-fault* emphasizes error overlap, *Q-statistic* emphasizes correlation, and *Entropy* emphasizes overall variety.

## 3 Method

### 3.1 Implicit Ensemble Framework

In order to investigate the effects of diversity in deep neural networks, we view a *single* neural network as an implicit ensemble. In particular, we adopt the framework introduced by Davel et al. [15], which treats each hidden node as a weak classifier. We refer to such a node classifier as a *subpredictor*.

Intuitively, hidden nodes tend to specialize during training. That is to say, they become sensitive to certain patterns or classes. Following Davel et al. [15], we estimate, for each node $n$ and class $c$, the class-conditional probability $P(z_n(x^{(i)}) \mid c)$ of each node's pre-activation $z_n$ given a sample $x^{(i)}$ by applying a kernel density estimator (KDE) trained using all $N$ training samples' activation values observed at the node.

Applying Bayes' rule yields a node-level posterior over classes for an input with pre-activation $z_n$:

$$P(c \mid z_n) = \frac{P(z_n \mid c) P(c)}{\sum_{c'} P(z_n \mid c') P(c')}, \quad (7)$$

where $P(c)$ is the class prior[1]. $P(c \mid z_n)$ is then node $n$'s probability for class $c$ when given a certain input. Then, for a given pre-activation $z_n(x^{(i)})$ and a node $n$ in layer $\ell$, the node's class prediction is computed as $\hat{y}_{\ell,n}(x^{(i)}) = \arg\max_c P(c \mid z_n(x^{(i)}))$. We refer to this as a *hard vote*.

**Majority vote ensemble:** To ensemble the output's of the subpredictors in a layer $\ell$, we employ a simple majority vote rule to determine our layer-wise prediction, and by extension, layer-wise accuracy. These hard votes are then used for computing the diversity of the predictions across nodes within the given layer.

**Nodes as subpredictors:** We first demonstrate that individual nodes can indeed be treated as meaningful subpredictors within the implicit ensemble. To do this, we compute the accuracy of each node when predicting class labels. Figure 2 shows a heatmap of node accuracies across layers, while Figure 3 presents the layer-wise accuracy for the train, validation, and test sets. These results confirm that individual nodes can perform as subpredictors.

Although the predictive *ability* of individual nodes within a neural network layer may be limited, their performance is consistently better than random chance. When the output of these subpredictors are aggregated, their collective predictions at the layer level yields better accuracies over the training, validation, and testing sets, similar in fashion to the principles of ensemble learning, where multiple 'weak learners' are combined to create a single 'strong learner.'

### 3.2 Applying Diversity Metrics

We focus on the diversity metrics mentioned in Section 2.2.

We compute these metrics directly from the node-level predictions. For pairwise measures like *Q-statistic*, *Disagreement*, or *Double-fault*, we calculate

---

[1]Class priors are taken as the empirical class frequencies in the training data.

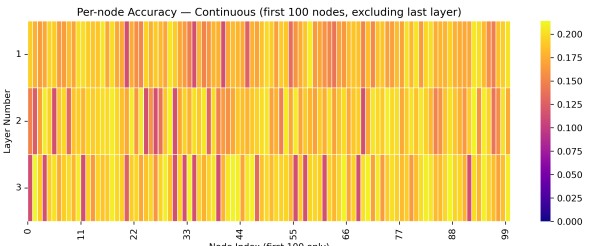

**Figure 2.** Accuracy heatmap of node predictions in a three layer MLP. Only the first 100 nodes are shown here. Most nodes in each layer have an individual accuracy of around 15-20% (higher than random guessing). Results were obtained from the MNIST test set for a model trained on 2,000 samples of the MNIST dataset.

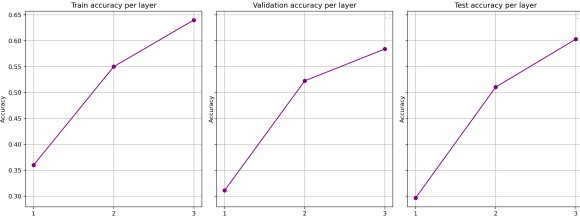

**Figure 3.** Layer-wise accuracy graphs for the train, validation, and test on MNIST. Accuracy improves with network depth. Each layer accuracy has significantly higher accuracy than compared to the individual node accuracies. This clearly showcases the ensembling strength of individual nodes within a layer.

the average over all unique pairs of nodes within a specific layer. For our non-pairwise metric i.e., *Entropy*, we compute the measures across all nodes in the layer directly.

To facilitate these calculations, we first needed to construct our *contingency tables* (see Table 1) for each combination of subpredictors within a layer. This was done by first creating a binary accuracy matrix, **B**, from our subpredictor predictions. Each entry, $\mathbf{B}_n^{(x^{(i)})}$, indicates whether node $n$ correctly classified sample $x^{(i)}$ by comparing the sample's true class label $y^{(i)}$ to the node's predicted class label $\hat{y}_{\ell,n}(x^{(i)})$:

$$\mathbf{B}_n^{(x^{(i)})} = \begin{cases} 1, & \text{if } \hat{y}_{\ell,n}(x^{(i)}) = y^{(i)} \\ 0, & \text{otherwise} \end{cases} \quad (8)$$

We then use the binary accuracy matrix to construct the contingency table for each pair of nodes $(n_1, n_2)$. This table records the number of samples for which both nodes made the same or different predictions, based on whether their predictions were correct (1) or incorrect (0). Then using the equations given in Section 2.2, we can compute each diversity metric across layers.

## 3.3 Training protocol

Due to how easily MLPs learn datasets like MNIST and Fashion-MNIST (with evaluation accuracies often being above 98%), and to more clearly see the effects of inducing diversity, we train the MLP networks on a small stratified subset of the training data (e.g., 2 000 samples) until interpolation (100% training accuracy). This protocol ensures that models generalize to different degrees despite being perfectly fit to the training subset. CNN models are trained using the full CIFAR-10 training set for all experiments besides the varying batch size experiments where a stratified subset of 8 000 samples was used. Models are trained using cross-entropy loss and optimized with the Adam optimizer, and no batch normalization has been applied. All experiments are repeated across five different random model seeds. We report the mean ± standard deviation for most of our results.

Details regarding datasets used, hyperparameter setup, and model architectures can be found in Appendix A.

## 3.4 Inducing diversity through regularization

Regularization methods are commonly used to improve generalization in deep learning [5]. To study the effect of induced diversity, we apply common regularization techniques that we believe should have an effect on the diversity within the network:

- **Dropout:** [4] randomly deactivates nodes during training, forcing different subsets of nodes to specialize. Applied to hidden layers with varying rates (0.1-0.5 for MLPs, 0.1-0.3 for CNNs).
- **DropConnect:** [20] similar to Dropout, except the weights are randomly removed rather than the activations. The same rates are applied as for Dropout.
- **Batch size variation:** [21] smaller batch sizes increase gradient noise which could possibly introduce some diversity to the nodes during training. Models are trained with batch sizes of: 16, 32, 64, 128, 256 (baseline), 512.

**Expected effects of induction:** For *Dropout* and *DropConnect*, increasing the drop probability $p$ should raise the node-level diversity by discouraging co-adaptation and encouraging specialization across stochastic subnetworks [4]. For *batch size*, smaller batches are expected to produce higher diversity because they introduce more gradient noise and more parameter updates per epoch, possible promoting more node-level diversity.

In this work, we use these techniques to explicitly manipulate diversity and investigate its effect on generalization in implicit ensembles.

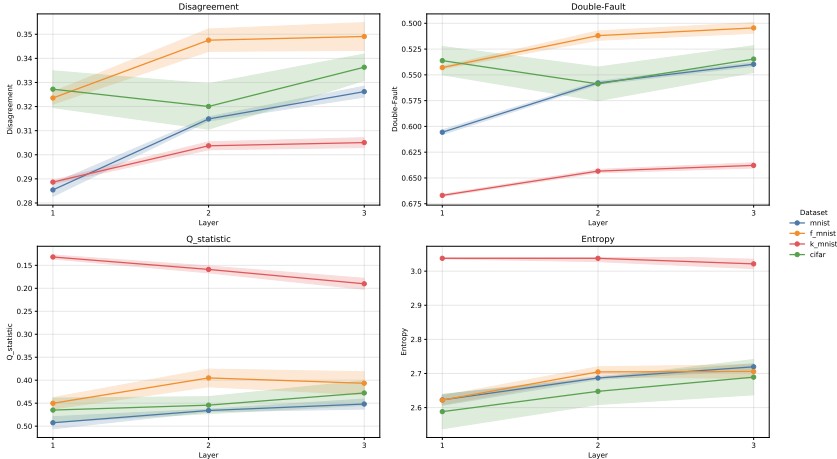

**Figure 4.** Mean diversity (averaged over model seeds) for *Disagreement*, *Double-fault*, *Q-statistic*, and *Entropy*, computed per layer for MNIST, K-MNIST, Fashion-MNIST (MLPs) and CIFAR-10 (CNN). Note that the y-axis for *Double-fault* and *Q-statistic* have been inverted in order to show increasing diversity.

# 4 Empirical Results

## 4.1 Diversity trends (no induced regularization)

Having established that nodes can act as subpredictors, we evaluate whether classical diversity measures provide meaningful insights in this implicit setting.

**Diversity relationship with depth:** Across the three-layer MLPs, mean diversity (averaged over five model seeds) increases monotonically with depth for most metrics. As shown in Figure 4, *Disagreement* and *Double-fault* exhibit complementary monotonic trends—*Disagreement* increases and *Double-fault* decreases from Layer 1 to 3 across MNIST, K-MNIST, and Fashion-MNIST—indicating higher diversity in deeper layers (note that for *Double-fault* and *Q-statistic*, we have inverted the y-axis to indicate increasing diversity). The *Q-statistic* and *Entropy* measures show more dataset dependent behavior: for MNIST and Fashion-MNIST, diversity increases across layers, while for K-MNIST it decreases. For CIFAR-10 (CNN), diversity first decreases then increases for *Disagreement* and *Double-fault*, while *Q-statistic* and *Entropy* increase across all three layers. This non-monotonic pattern in the CNN (for *Disagreement* and *Double-fault*) likely reflect the distinct role of the fully-connected stage in the convolutional network, where its function differs from the hierarchical transformations in MLPs.

**Layer accuracy correlated with diversity:** Layer accuracies also increase with depth for all MLP models, confirming that deeper layers yield more discriminative subpredictors. Figure 5 shows that *Disagreement* and *Double-fault* are strongly correlated with layer accuracy across datasets (Pearson $r > 0.85$ for all MLPs), demonstrating that diversity mirrors the layer performance. In contrast, *Q-statistic* and *Entropy* display dataset-dependent correlations—for MNIST and Fashion-MNIST, we have a strong negative correlation with *Q-statistic*, and a strong positive correlation with *Entropy*, but weak or even opposite for K-MNIST ($r = 0.500$ with *Q-statistic*, and $r = 0.029$ with *Entropy*). For CIFAR-10, we see similar trends to MNIST and Fashion-MNIST although weaker in correlation strength.

**Summary:** Across datasets, *Disagreement* and *Double-fault* show the most reliable and interpretable correlations between layer diversity and layer performance. Layers with higher *Disagreement* (and lower *Double-fault*) consistently achieve higher accuracy, supporting the link between diversity and predictive strength. In contrast, *Q-statistic* and *Entropy* exhibit dataset-dependent or inconsistent behavior—particularly for K-MNIST—where correlations vary in both magnitude and direction. These two measures therefore provide limited cross-dataset generality and are treated as secondary in our analyses.

## 4.2 Inducing Diversity

**Regularization effects on model performance:** We investigate how certain regularization techniques—Dropout, DropConnect, and batch size—affect model generalization and diversity.

Figure 6 shows test accuracy against Dropout probability across datasets. Increasing the drop rate $p$ improves model accuracy for all datasets. For CIFAR-10, once $15 - 20\%$ Dropout is applied, performance begins to plateau or decline slightly,

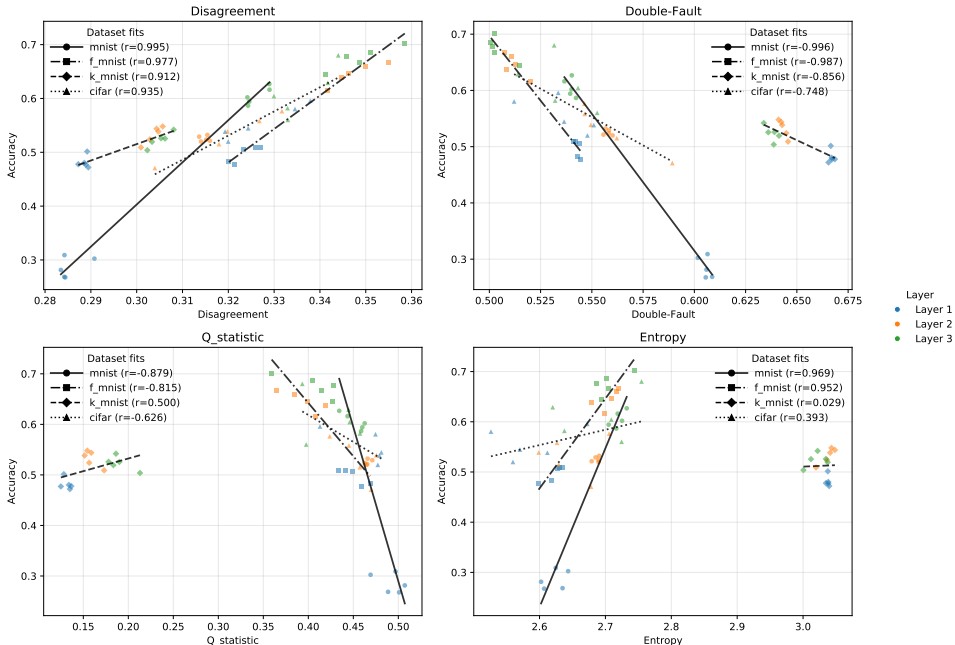

**Figure 5.** Scatterplots of layer-performance versus layer-diversity for all datasets and metrics, indicating the correlation between diversity and layer accuracy.

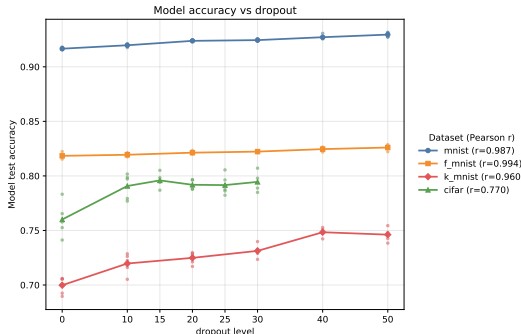

**Figure 6.** Model test accuracy versus Dropout probability across datasets. Model performance improves consistently with increasing Dropout.

indicating a possible onset of over-regularization. Similar behavior is observed for DropConnect and when reducing the training batch size, where smaller batches yield higher test accuracy. These results confirm moderate regularization improves model performance across all datasets. DropConnect and batch size results can be seen in Figures B.1 and B.2, respectively.

**Regularization effects on diversity:** To assess how regularization influences diversity within the network, we measure layer-wise diversity at varying Dropout probabilities for Fashion-MNIST (Figure 7). Each point represents a model initialized with a specific seed and then trained with a specific Dropout level. Across layers, observe a positive correlation between layer accuracy and *Disagreement*—$r = 0.639$ for Layer 1, $r = 0.661$ for Layer 2, and $r = 0.909$ for

Layer 3—indicating that higher Dropout promotes both increased diversity and improved predictive performance. The relationship strengthens in deeper layers.

Averaging across layers, the same pattern holds across datasets (Figure 8). *Disagreement* increases and *Double-fault* decreases with stronger regularization, reflecting greater diversity among subpredictors. In contrast *Q-statistic* still exhibits dataset-dependent trends—K-MNIST still diverges from metrics—while *Entropy* generally decreases with Dropout, showing only minor fluctuations. See Figures B.3 and B.4 for results obtained for DropConnect and batch size.

**The link between generalization and diversity:** To evaluate whether induced diversity correlates with generalization, we compute the Pearson correlation between model test accuracy and average layer diversity across all regularization settings and datasets. Figure 9 summarizes these relationships for the four diversity measures when applying Dropout to the network.

Across datasets and regularizers, increasing the strength of the regularizer—through higher drop probabilities $p$ for Dropout and DropConnect, or smaller batch sizes—yields both higher test accuracy and higher average diversity, although the strength of this link depends on the metric and the dataset. For the *Disagreement* measure, correlations are consistently positive and strong ($r = 0.629$-$0.932$ for Dropout), confirming that greater node-level disagreement coincides with better model

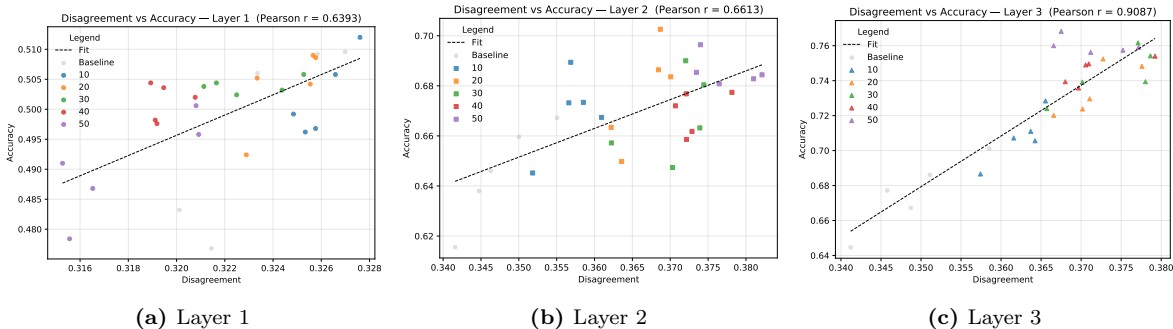

**Figure 7.** Scatterplots of layer performance versus layer *Disagreement* for Fashion-MNIST under different Dropout probabilities. Each point represents a model trained with a specific Dropout rate.

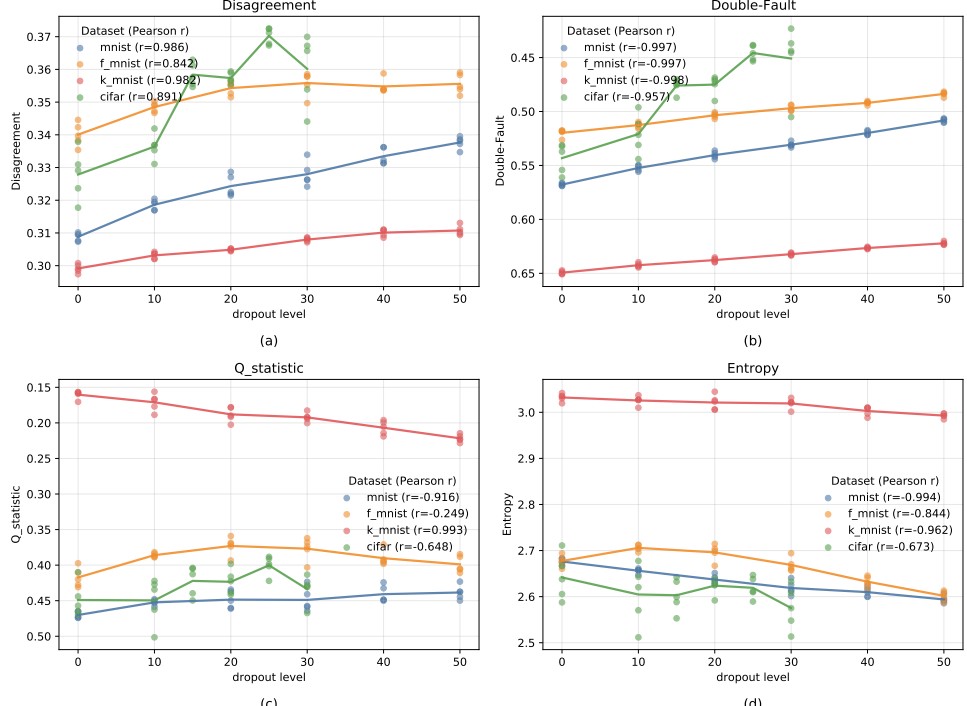

**Figure 8.** Average diversity versus Dropout probability across datasets for all diversity metrics. Note that the y-axis for *Double-fault* and *Q-statistic* have been inverted in order to show increasing diversity.

performance. The complementary *Double-fault* measures shows equally strong but negative correlations ($r = -0.622\text{-}-0.953$), as lower shared-error frequency indicates higher diversity and thus improved generalization.

In contrast, *Q-statistic* and *Entropy* display less consistent behavior. The *Q-statistic* reverses sign for K-MNIST ($r = 0.805$), suggesting dataset-specific dependencies in how subpredictor agreement relates to performance.

*Entropy* is largely negative across datasets (e.g., MNIST $r = -0.932$). However, the CIFAR-10 case shows small, near-zero or even positive correlations (Figure B.6), which could suggest that for convolutional models, the distributional spread of node predictions behave differently from the fully connected MLPs.

Overall, these results demonstrate a clear positive association between generalization and diversity: models that generalize better tend to exhibit greater diversity among subpredictors. The relationship is most consistently and robustly captured by the *Disagreement* and *Double-fault* measures, while *Q-statistic* and *Entropy* depend more strongly on dataset and architecture.

**Further results:** Additional results for the remaining regularization methods (DropConnect and batch size) and their effects on model performance, diversity, and the relationship between the two, are provided in the supplementary material.

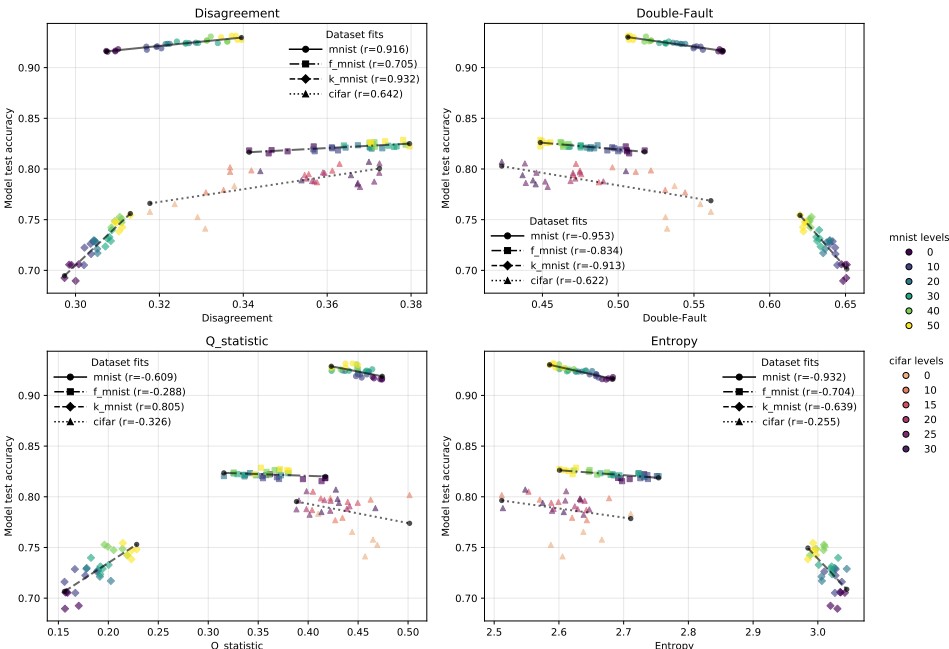

**Figure 9.** Correlation between model test accuracy and diversity across datasets and metrics under induced-diversity conditions using Dropout.

## 5 Conclusion

This work examined how diversity within over-parameterized neural networks relates to generalization by treating each hidden node as a subpredictor of an *implicit ensemble*. Using classical ensemble diversity measures, we quantified layer-wise diversity across multiple datasets and architectures and investigated how this diversity behaves both without and with regularization such as Dropout, DropConnect, and batch sizes.

Our results show that diversity generally increases with depth. Additionally, we find that regularizers such as Dropout, DropConnect, or smaller batch sizes, generally increases the diversity in a neural network's layers.

We find that layer diversity strongly correlates with layer accuracy. When regularizers are included (Dropout, DropConnect, and batch size), both layer accuracy and diversity generally increase. Among all measures, *Disagreement* and *Double-fault* emerge as the most consistent indicators of this correlation—showing strong positive and negative correlations with accuracy, respectively. In contrast, *Q-statistic* and *Entropy* exhibit dataset- and architecture-dependent variability.

Finally, with respect to a neural network's generalization, we observed that diversity within a single neural network, averaged over layers, is correlated with model test accuracy.

**Future work:** As a first step, this work provides an exploratory study of the relationship between a neural network's diversity and generalization. Towards a next step in this study, an interesting direction would be to extend this framework to larger and more modern architectures such as ResNets and Transformers, potentially using pre-trained large-scale models (e.g., ImageNet networks trained with Dropout) to study diversity at scale. Another valuable next step would be to directly compare the diversity-performance relationship in neural networks to that of *explicit ensembles*.

There are also intriguing questions that emerge from this study. Notably, does a tradeoff between a neural network's diversity and generalization exist? For example, in this investigation, we used a moderate amount of regularization. As the strength of the regularizer is increased, we hypothesize a decrease in both the model's generalization performance and diversity. Finally, there are also discrepancies between different diversity measures, particularly the divergent behavior of the *Q-statistic* and *Entropy*. An open problem is to better understand this discrepancy and to understand their sensitivity to datasets and architectures.

## Acknowledgements

This work is based on the research supported in part by the National Research Foundation of South Africa (Ref Numbers: PSTD23042898868, RCDL240215206999).

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

## A  Model setup

**Datasets:**  We evaluate our approach on four standard image classification benchmarks:

- MNIST [22], Fashion-MNIST [23], and K-MNIST [24] grayscale digit and clothing classification datasets with 10 classes. K-MNIST is a variant of MNIST with Japanese characters.
- CIFAR10 [25], a colored natural image dataset with 10 classes.

**Architectures:**  We used different architectures to confirm our results.

- For MNIST, Fashion-MNIST, and K-MNIST, we use fully-connected multi-layered perceptrons (MLPs) with a depth of three hidden layers, each with a width of 512 nodes.
- For CIFAR-10, we use a convolutional neural network (CNN) with multiple convolutional and pooling layers followed by fully connected layers (see Table A.2 for details). Diversity measures are applied only to the fully connected layers, where subpredictors are defined at the node level.

**Hyperparameter setup:**  The specific details regarding what hyperparameters were used to train each model can be found in Table A.1. All models were trained with a baseline batch size of 256, while the varying batch size experiments then used the other values. No Dropout or DropConnect was

**Table A.1.** Hyperparameter setup for different model architectures

| Hyperparameter | Value |
|---|---|
| Optimizer | Adam |
| Learning rate | 0.0003 |
| Batch size | {16, 32, 64, 128, 256 (baseline), 512} |
| Epochs (max) | 1000 |
| Dropout/DropConnect rates (MLPs) | {0, 0.1, 0.2, 0.3, 0.4, 0.5} |
| Dropout/DropConnect rates (CNNs) | {0, 0.1, 0.15, 0.2, 0.25, 0.3} |
| Learning rate scheduler | {Step size = 1, gamma = 0.99} |

applied to normal baseline models. A learning rate scheduler was also applied for all different models.

The specific model architecture for the CNNs can be found in Table A.2.

**Table A.2.** CNN specification (conv bias=True, activation=ReLU, BN=False). "$\times n$" indicates repeated blocks.

| Stage | Layers | Kernel / Stride / Pad | Output size (for 32×32 input) |
|---|---|---|---|
| Input | | | 32×32×3 |
| S1 | Conv 64 ×2 | (3, 1, 1) | 32×32×64 |
| | MaxPool | (2, 2, 0) | 16×16×64 |
| S2 | Conv 128 ×2 | (3, 1, 1) | 16×16×128 |
| | MaxPool | (2, 2, 0) | 8×8×128 |
| S3 | Conv 256 ×3 | (3, 1, 1) | 8×8×256 |
| | MaxPool | (2, 2, 0) | 4×4×256 |
| S4 | Conv 512 ×3 | (3, 1, 1) | 4×4×512 |
| | MaxPool | (2, 2, 0) | 2×2×512 |
| S5 | Conv 512 ×3 | (3, 1, 1) | 2×2×512 |
| | MaxPool | (2, 2, 0) | 1×1×512 |
| Head | Flatten → MLP(3×512) → Dense($C$) | | $C$ |

## B  Additional results

**Regularization effects on model performance:** Figures B.1 and B.2 illustrate the effects that DropConnect and varying batch size have on the model accuracy. We observe that increasing the amount of DropConnect or decreasing the batch size yields a corresponding improvement in model performance across all datasets. This behavior is consistent with prior literature, which highlights regularization as a mechanism that generally enhances model generalization.

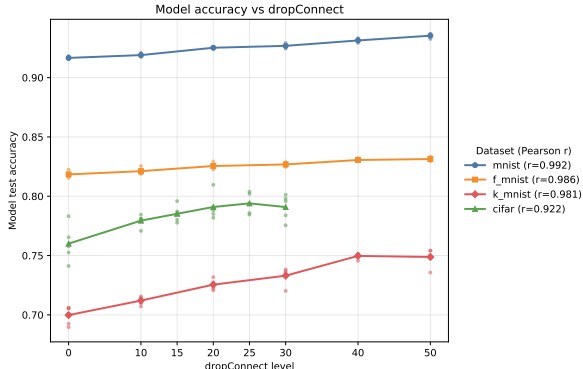

**Figure B.1.** Model test performance against percentage DropConnect applied during training across datasets. There is a clear increasing trend in accuracy as the drop probability being applied is increased.

**Regularization effects on diversity:** Figures B.3 and B.4 illustrate the effects that DropConnect and varying batch size have on the diversity

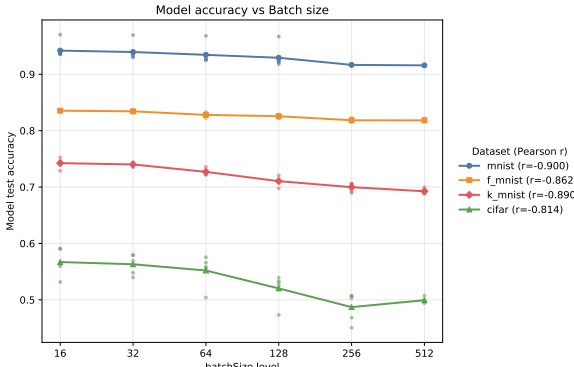

**Figure B.2.** Model test performance for models trained with varying batch sizes. There is a clear in accuracy with smaller batch sizes.

within the network. We observe that increasing the amount of DropConnect or decreasing the batch size generally yields a corresponding increase in diversity across all datasets—this is evident by the clear correlation we see between diversity and how much DropConnect is applied or the size of the batches. We do however, note some inconsistencies with some metrics, where the Q-statistic and Entropy measures show some contradictory results—especially on the K-MNIST dataset. The results suggest—particularly for the *Disagreement* and *Double-fault* measures—that added DropConnect and smaller batch sizes generally lead to increased diversity.

**Relationship between diversity and generalization performance:** Figures B.5 and B.6 illustrate the effects that DropConnect and varying batch size have on the relationship between diversity and generalization performance. We generally observe strong positive relationships between increased diversity and improved generalization. This is particularly evident for the *Disagreement* and *Double-fault* measures, both of which show strong correlations with model performance in the DropConnect and batch size experiments. For DropConnect, we obtain Pearson correlations of $r > 0.71$ for *Disagreement*, and $r < -0.79$ for *Double-fault*, while decreasing the batch size yields similarly strong trends with $r > 0.55$ and $r < -0.67$, respectively. We still find some inconsistencies with the results for the *Q-statistic* and *Entropy* measures, where they show weaker or even reversed trends for some datasets (especially the K-MNIST dataset). Besides these inconsistencies, we do however, still see some trend where an increase in diversity correlates with improved diversity.

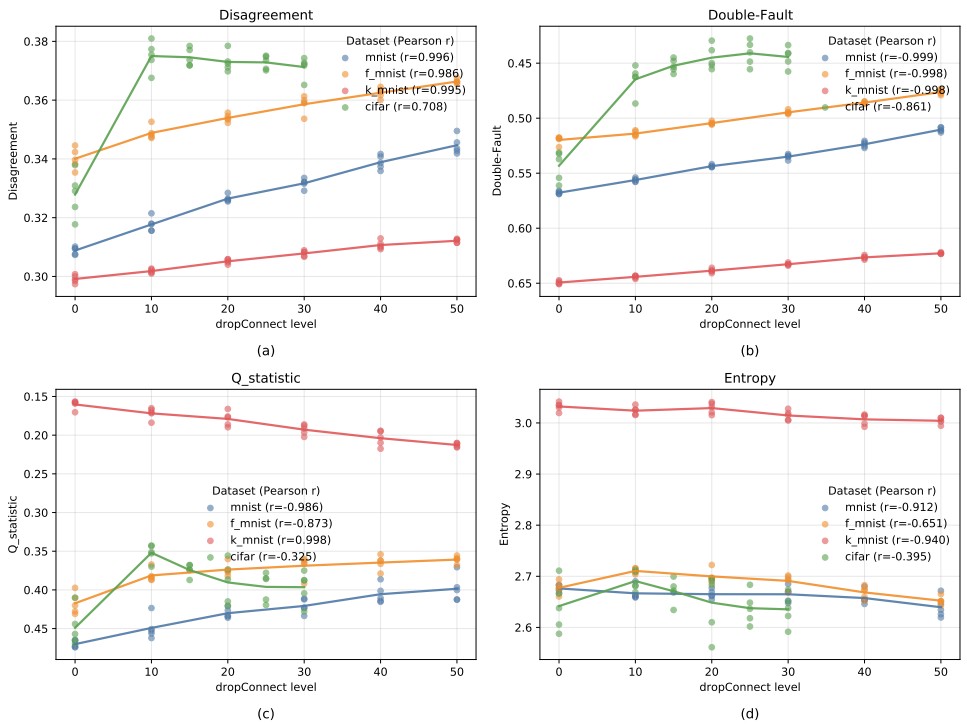

**Figure B.3.** Average diversity versus DropConnect probability across datasets for all diversity metrics. *Disagreement* increases and *Double-fault* decreases with DropConnect level, indicating higher diversity at stronger regularization. *Q-statistic* shows mixed behavior, with K-MNIST diverging from other datasets, while *Entropy* decreases slightly with minor fluctuations.

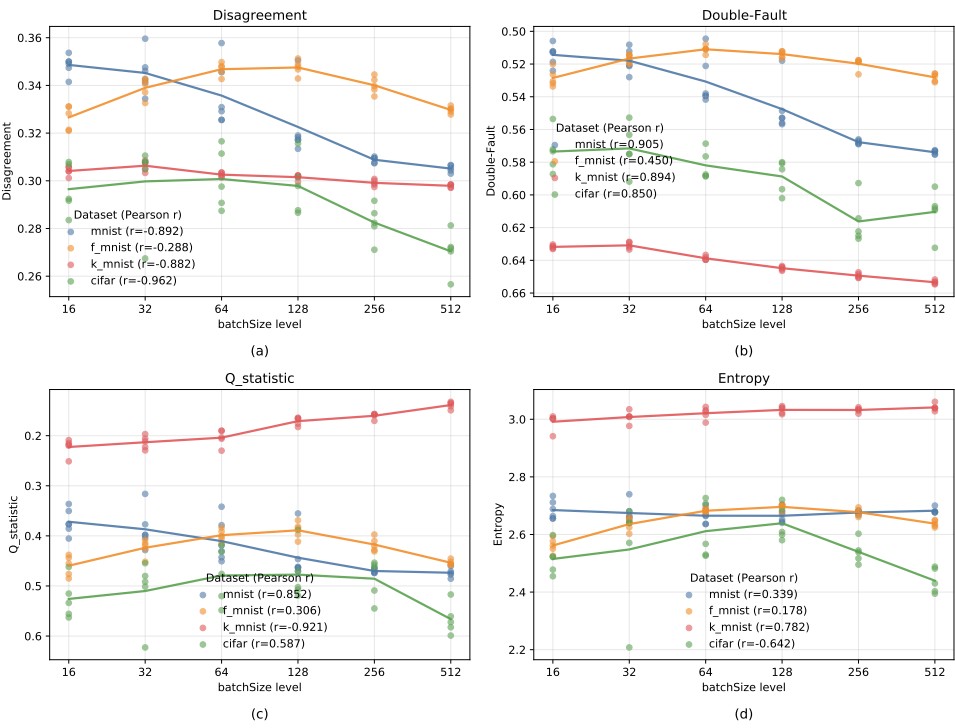

**Figure B.4.** Average diversity versus batch size across datasets for all diversity metrics. *Disagreement* generally increases and *Double-fault* generally decreases with smaller batch sizes, indicating higher diversity. *Q-statistic* and *Entropy* show mixed behavior, with K-MNIST and CIFAR-10 (only for the *Entropy*) diverging from other datasets.

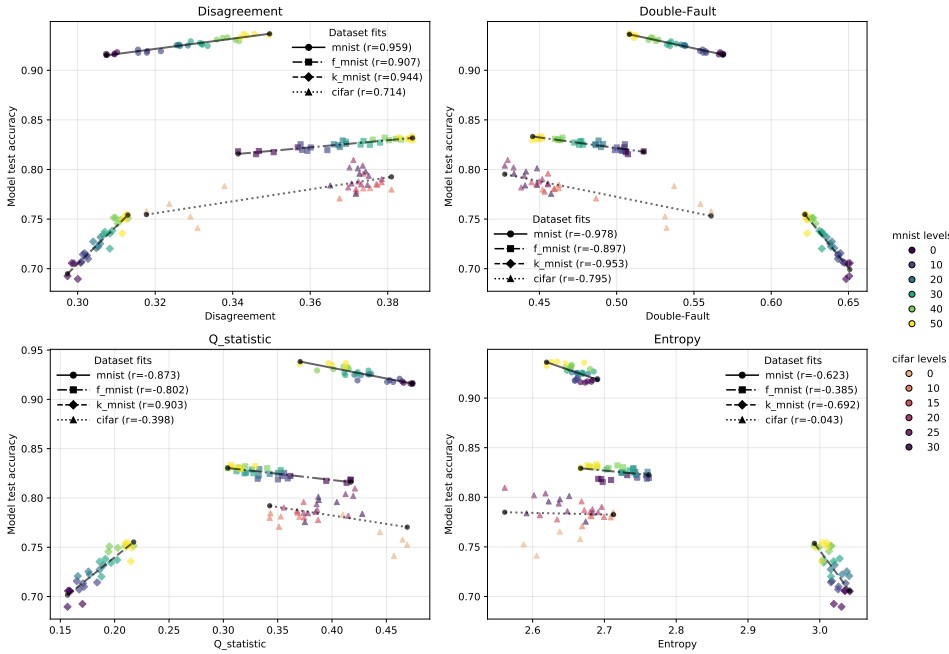

**Figure B.5.** Correlation between model test accuracy and averaged diversity across datasets and metrics under induced diversity conditions using DropConnect.

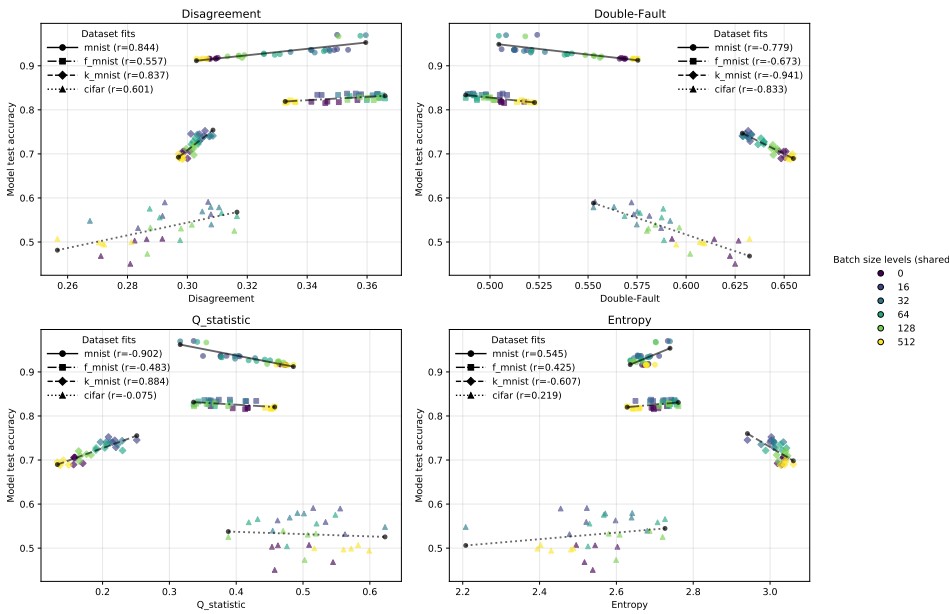

**Figure B.6.** Correlation between model test accuracy and averaged diversity across datasets and metrics under induced diversity conditions using varying batch sizes.

