# OpenReview forum: "Investigating the relationship between diversity and generalization in over-parameterized deep neural networks"
_NLDL.org/2026/Conference — NLDL 2026 Poster_

### Official Review · Reviewer_p4MW · 2025-09-25
**Investigating the relationship between diversity and generalization in deep neural networks**

**Rating:** 4
**Confidence:** 4
**Final Rating:** 4
**Final Confidence:** 4

**Summary:**

The paper examines the relation between diversity amongst neurons in a neural network and generalization performance in the interpolation regime (zero training error). The methods is centered on supervised classification, and diversity is measured using four different metrics, which are based on treating each neuron as a (weak) classifier. These weak classifiers are constructed as Parzen/KDE Bayes classifiers. It is shown empirically how the diversity increases through the layers of a small MLP. The paper then examines how diversity is affected by regularization techniques (dropout and dropconnect) as well as batch size, and how this correlates with test error.

**Strengths:**

Understanding generalization in neural networks is a very interesting topic.

The paper is well written and clearly structured.

Enough technical details are included, that I feel confident that results could be reproduced. Methods are clearly explained, and experiments are simple and direct.

**Weaknesses:**

The contribution is not strongly novel, and does not lead to strong actionable conclusions. I think this is a very interesting line of work, but I am not sure if we can conclude much.

Results could be more clearly presented. For example, results for several of the settings and metrics are not included in the main paper. The results in tables in the supplement would be more clearly communicated by graphs (but fine to have the tables in addition.)

For the Fashion-MNIST dataset, the dropout rates and batch sizes considered only lead to improved performance. It would be interesting also to see what happens for example when dropout rates are set "too high" - then accuracy would drop, but what would happen with the diversity metrics? I assume also that this might take us out of the interpolation regime, so that could complicate matters.

Questions and recommendations:

- Why do the result for the Q-statistic and especially the entropy not follow the same trend as for Diversity and Double Fault?

- If I understand it correctly, the "Disagreement" metric more accurately measures differences in correctness, since classifiers could also disagree while both being wrong in the multiclass setting. If that is the case, the wording around the definition of Disagreement should be adjusted.

- Why are the accuracies for CIFAR-10 Batch size in table C.2 so low? Should they not match the 0% Dropout/DropConnect numbers at the baseline batch size?

- Similar to table C.1 and C.2, do you not have results for MNIST and K-MNIST?

- Some interesting results are only given in the appendix - consider moving more results to the main paper (e.g. B.1, C.1, and C.2) possibly converting to graphs while keeping tables in the supplement. It would be nice to clearly show how diversity increases/changes through layers for all four datasets and all four metrics. Similarly, show test accuracy and disagreement metrics for all data sets and all three regularization methods.

- I wonder if the title would be better if it mentioned that we are in the interpolation regime, e.g. "... in overparameterized neural networks". Similarly, it could be mentioned in the abstract.

**Final Justification:**

Promising early-stage research offering interesting observations

**Justification:**

I am inclined to recommend accepting the paper, but I strongly suggest improving the presentation to ensure that all results are clearly and readably conveyed in the main text. The paper explores the relationship between generalization and diversity, and demonstrating this more clearly across datasets, models, and metrics would increase its impact. While some of the conclusions are somewhat vague, the topic is engaging and could stimulate interesting discussions at the conference.

---

> ### Author Rebuttal · Authors · 2025-10-21
>
> We thank the reviewer for the constructive feedback. We appreciate the positive remarks regarding the clarity and structure of the paper, the reproducibility of the experimental setup, and the recognition that understanding generalization in neural networks is an important direction of study.
>
> Below we address the concerns raised, as well as the questions:
> # Weaknesses:
>
> >The contribution is not strongly novel, and does not lead to strong actionable conclusions.
>
> **Response (1)**: Our work is positioned as an initial exploratory study; in particular, through viewing a neural network as an implicit ensemble, we are able to study the relationship between various diversity measures, constructed for explicit ensembles, and a neural network’s generalization. To the best of our knowledge, such a study has not been previously performed. Kindly see also our Response (2) to Reviewer q5rq.
>
> >Results could be more clearly presented. For example, results for several of the settings and metrics are not included in the main paper. The results in tables in the supplement would be more clearly communicated by graphs (but fine to have the tables in addition.)
>
> **Response (2)**: Due to the 5-page submission limit, we prioritized clarity and avoided overcrowding the main paper with multiple graphs, opting for concise tables and full metric breakdowns to the supplement. We agree that visualization could further enhance interpretability, and in a longer version or future work we would like to include graphical summaries alongside tables for improved clarity.
>
> >For the Fashion-MNIST dataset, the dropout rates and batch sizes considered only lead to improved performance. It would be interesting also to see what happens for example when dropout rates are set "too high" - then accuracy would drop, but what would happen with the diversity metrics? I assume also that this might take us out of the interpolation regime, so that could complicate matters.
>
> **Response (3)**: We agree that exploring over-regularized settings (e.g., high dropout) would provide a valuable contrast in understanding diversity behaviour. In fact, we experimented with higher dropout levels in preliminary tests to probe this trade-off, but found that once dropout exceeded roughly 60% on the Fashion-MNIST dataset for example, models began to leave the interpolation regime. For this reason, we focused the present study on practically relevant regularization ranges where models remain in the interpolation regime.
>
> We do, however, observe early signs of this “too much dropout” effect in our CIFAR-10 results, where diversity continues to increase slightly while accuracy starts to decline, hinting at the onset of this trade-off.
> # Questions and recommendations:
> >Why do the result for the Q-statistic and especially the entropy not follow the same trend as for Diversity and Double Fault?
>
> **Response (4)**: We find this to be one of the most intriguing observations in our results. As this work is positioned as exploratory, our focus was on establishing the measurement framework and reporting emerging patterns. We call attention to this divergence as a key area for future investigation as it may indicate that some diversity measures capture qualitatively different interaction behaviours within implicit ensembles compared to others such as Disagreement and Double-fault.
>
> >If I understand it correctly, the "Disagreement" metric more accurately measures differences in correctness, since classifiers could also disagree while both being wrong in the multiclass setting. If that is the case, the wording around the definition of Disagreement should be adjusted.
>
> **Response (5)**: We agree that the wording could lead to ambiguity in the multiclass case, and we will refine the description of the Disagreement metric to make this interpretation clearer in the paper.
>
> >Why are the accuracies for CIFAR-10 Batch size in table C.2 so low? Should they not match the 0% Dropout/DropConnect numbers at the baseline batch size?
>
> **Response (6)**: Thank you for mentioning this. The lower baseline accuracy in Table C.2 is due to the fact that, unlike the dropout and dropconnect experiments which were trained on the full CIFAR-10 dataset, the batch size experiments were trained on a stratified subset of 8,000 samples. We will make this distinction clearer in the training protocol section.
>
> >Similar to table C.1 and C.2, do you not have results for MNIST and K-MNIST?
>
> **Response (7)**: Yes, we do have corresponding results for MNIST and K-MNIST following the same protocol used for tables C.1 and C.2. These were omitted from the current supplementary material for space and redundancy reasons, as we observed consistent trends across all the datasets. We will add these to the supplemental material for further clarity.
>
> >Some interesting results are only given in the appendix - consider moving more results to the main paper (e.g. B.1, C.1, and C.2) possibly converting to graphs while keeping tables in the supplement. It would be nice to clearly show how diversity increases/changes through layers for all four datasets and all four metrics. Similarly, show test accuracy and disagreement metrics for all data sets and all three regularization methods.
>
> **Response (8)**: Thank you for this suggestion. In the initial 5-page submission format, we prioritized concise core results and moved the full cross-dataset tables to the appendix to avoid overcrowding. However, if accepted, the extended page-limit will give us the space to move selected appendix results into the main paper as figures. We agree that showing diversity progression across layers for all datasets and metrics in compact plots would strengthen the narrative, and we intend to include these visualizations in the camera-ready version.
>
> >I wonder if the title would be better if it mentioned that we are in the interpolation regime, e.g. "... in overparameterized neural networks". Similarly, it could be mentioned in the abstract.
>
> **Response (9)**: We agree that explicitly referencing the interpolation/overparameterized regime in the title or abstract would make the experimental context clearer and more accurately give the setting of our analysis. We have revised the title and abstract wording accordingly.

---

### Official Review · Reviewer_q5rq · 2025-10-07
**Weaknesses outweigh strengths**

**Rating:** 2
**Confidence:** 3
**Final Rating:** 2
**Final Confidence:** 3

**Summary:**

This paper views neural networks as an emsemble, where each node acts as a predictor, and presents an empirical study aiming to gain insights into how the diversity of the individual node's prediction capability is related to the generalization ability of the overall network.

**Strengths:**

The paper investigates an interesting viewpoint of neural networks as emsembles and presents the empirical analysis clearly with sufficient background and expressive figures.

In Fig 1, a trend that seems to be visible is that increased DropConnect values reduces the variance of the model. Could the authors share any intuition they may have on why this is so?

**Weaknesses:**

Neither the methods used nor the insights seem novel. For example, it is already known that regularization in neural networks plays a similar role to diversity control in ensembles in terms of generalization [1]. It is also already known that feature specialization class discrimination improves with layer depth [2].

The study is carried on a small simple 3-layer MLP. Perhaps a deeper model, or the experiments conducted for a variety of machine learning tasks such as graph or language in addition to vision tasks, could yeild newer insights.

[1] Diversity Regularized Machine. (IJCAI 2011)
[2] Visualizing and Understanding Convolutional Networks (ECCV 2014)

**Final Justification:**

While I accept the author's point that there is a novelty factor of studying the diversity of a single network, I still believe the study is too limited. Even as an exploratory study, more insights can be incorporated, as suggested by reviewer LjMm, to make the study more convincing and comprehensive.

**Justification:**

The study seems too preliminary and non-conclusive with the key takeaways being already known.

---

> ### Author Rebuttal · Authors · 2025-10-21
>
> We thank the reviewer for the feedback. We appreciate the recognition of the implicit ensemble viewpoint taken in this work, as well as the acknowledgement that the empirical analysis, figures, and supporting background were clearly presented.
>
> Below we address the concerns raised, as well as the questions:
>
> # Questions:
> >In Fig 1, a trend that seems to be visible is that increased DropConnect values reduces the variance of the model. Could the authors share any intuition they may have on why this is so?
>
> **Response (1)**: We do notice the reduction in variance as DropConnect increases, but at this stage, we do not have a strong intuition for why this occurs. We view this as an interesting direction for further analysis beyond the scope of this exploratory study.
> # Weaknesses:
> >Neither the methods used nor the insights seem novel. For example, it is already known that regularization in neural networks plays a similar role to diversity control in ensembles in terms of generalization (Diversity Regularized Machine. (IJCAI 2011)). It is also already known that feature specialization class discrimination improves with layer depth (Visualizing and Understanding Convolutional Networks (ECCV 2014)).
>
> **Response (2)**: Our work is positioned as an initial exploratory study. While the role between diversity and generalization in ensembles has been studied, we note that:
>
> * Several different measures for diversity exist which means that diversity is not well-defined (however, see the recent work of Wood et al. [12]).
> * The role of diversity in neural networks is less well understood: to apply diversity metrics requires the notion of ensemble members. In neural networks, it is not completely clear what is a member (subpredictor) of the (implicit) ensemble (there are several different ways to define an implicit ensemble e.g., Veit et al. [13], Olson et al. [14], Davel et al. [15], and Benjamin et al. [16]).
> * With respect to their training, there are fundamental differences between neural networks and explicit ensembles. It is not clear how this impacts diversity of the subpredictors.
>
> Based on these points, we believe that our initial exploratory study contributes in the following sense:
> * By curating a selection of well-known ensemble diversity measures and by adopting a view of the neural network as an implicit ensemble (Davel et al. [15]), we are able to apply the ensemble diversity measures to a single neural network. To the best of our knowledge, such a study has not been previously performed for the case of a neural network.
> * While our paper largely confirms that increasing diversity tends to have a positive (regularizing) effect on a neural network’s generalization, we also observe fascinating differences in trends depending on which diversity measure is being used (see Table B.1 and Figure B.1).
>
>
> >The study is carried on a small simple 3-layer MLP. Perhaps a deeper model, or the experiments conducted for a variety of machine learning tasks such as graph or language in addition to vision tasks, could yield newer insights.
>
> **Response (3)**: While the main paper presents results for a 3-layer MLP on Fashion-MNIST, we also conducted experiments on a CNN model trained on CIFAR-10, with the results included in the appendix. The choice to highlight the MLP setting in the main body was intentional to clearly demonstrate the core results in a controlled setting with a more complex dataset than MNIST, before extending it to a more convolutional architecture. This work is positioned as exploratory and our computational budget constrained us from immediately scaling to architectures such as ResNets or Transformers. We explicitly state future work will extend this diversity analysis to modern large-scale architectures and datasets such as CIFAR-100 and ImageNet, where we believe the implicit ensemble interpretation may reveal even richer diversity dynamics. We agree that extending this analysis to other modalities such as language or graph models could reveal additional insights.

---

### Official Review · Reviewer_LjMm · 2025-10-08
**Some weaknesses, but overal accept**

**Rating:** 4
**Confidence:** 4

**Summary:**

The paper "Investigating the relationship between diversity and generalization in deep neural networks" investigates whether concepts from ensemble diversity can also help explain generalization within a single deep neural network (DNN). The authors treat a neural network as an implicit ensemble, where each hidden node acts as a weak “subpredictor.” Using this framework, they apply established ensemble diversity measures to study correlations between diversity and generalization across different layers, datasets, and regularization settings. More specifically, they study the disagreement, double-fault, Q-statistic, and entropy for models with dropout, dropconnect and for different batch sizes on four datasets. Findings are: Diversity generally increases with layer depth; Dropout and DropConnect are effective methods to increase diversity; Disagreement/Double-Fault seem to be the most reliable measures.

**Strengths:**

- Relevant questions and overall interesting idea to combine Davel et al. 2020 with the idea of diversity
- Well-executed experiments with clear empirical findings
- Overall simple, yet systematic approach leading to findings that can be built upon

**Weaknesses:**

- While experiments are well-executed, they are overall limited on smaller datasets and older model architectures.
- No comparison to explicit ensemble methods: The central claim is that “a single network can be viewed as an implicit ensemble.” However, the study does not compare this implicit ensemble’s diversity–performance relationship with actual, explicit ensembles (e.g. Deep Ensembles, Negative Correlation Learning, etc). The node-as-classifier assumption (c.f. Davel et al. 2020), is hence a bit debatable in this context. Individual node activations (e.g. on the first few layers) are not directly trained for classification, and hence it is expected that accuracy improves as we increase depth. Here, a direct comparison to explicit ensembles would strengthen the paper.

**Justification:**

It's overall a well-written paper with a well-executed study that is worth sharing. While limited, this paper can be grounds for future work.

In case this paper is rejected, I propose the following changes:

1) I understand that training large networks, and especially transformers, is resource-intensive and time-consuming. As a middle ground, I suggest a study of pre-trained models (e.g. for ImageNet) that are often trained with dropout. This way, the main claims of the paper can be further backed by an investigation of some large, "real-world" models. It should be easy to reproduce Fig {2,3,4} for pre-trained models as well

2) You should include some experiments with explicit ensembles. So far, you've focused on finding correlations between diversity and performance, but you did not provide an insight about the actual values or value ranges of diversity (i.e. is a diversity of 0.8 good? What does that even mean?). In the classical bias-(co-)variance decomposition, we explicitly know the trade-off between bias, (co-)variance (i.e., diversity), and the overall loss. Your approach currently lacks this. I suggest including experiments with explicit ensembles and comparing diversities here as a first step.

3) You can mention some additional literature, that might be valuable here, especially if you consider explicit ensembles:
[1] Generalized Negative Correlation Learning for Deep Ensembling by Buschjäger et al. 2020 (https://arxiv.org/abs/2011.02952)
[2] To Ensemble or Not Ensemble: When does End-To-End Training Fail? by Webb et al. 2020 (https://arxiv.org/abs/1902.04422)
[3] Learning with Pseudo-Ensembles by Bachman et al. 2014 (https://arxiv.org/abs/1412.4864)

---

> ### Author Rebuttal · Authors · 2025-10-21
>
> We thank the reviewer for the thoughtful comments and constructive feedback. We appreciate the positive recognition of the relevance of the research question, the combination of Davel et al. (2020) with diversity analysis, and the feedback noting that our approach is simple, systematic, and supported by clear empirical findings.
>
> Below we address the concerns raised, as well as the questions:
> # Weaknesses:
> >While experiments are well-executed, they are overall limited on smaller datasets and older model architectures.
>
> **Response (1)**: This work is intended as an exploratory first step, focusing on a simpler setting to demonstrate the diversity effects we study. Given limited computational resources, we started with smaller models, and as noted in the paper, we plan to extend this framework to modern architectures like ResNets and Transformers on larger datasets in future work.
>
> >No comparison to explicit ensemble methods: The central claim is that “a single network can be viewed as an implicit ensemble.” However, the study does not compare this implicit ensemble’s diversity–performance relationship with actual, explicit ensembles (e.g. Deep Ensembles, Negative Correlation Learning, etc). The node-as-classifier assumption (c.f. Davel et al. 2020), is hence a bit debatable in this context. Individual node activations (e.g. on the first few layers) are not directly trained for classification, and hence it is expected that accuracy improves as we increase depth. Here, a direct comparison to explicit ensembles would strengthen the paper.
>
> **Response (2)**: Our work follows in the same spirit as other works that have proposed an implicit ensemble picture of a neural network (e.g. Veit et al. [13], Davel et al. [15], and Benjamin et al. [16]). Notably, these works do not perform comparisons with explicit ensemble methods either. However, we do agree that a comparison of the diversity-performance of a neural network to explicit ensemble models would be valuable, especially since there are important differences in the training algorithms for a neural network and an explicit ensemble. We see this as a fascinating extension of our results for future work.
> # Recommendations:
> >I understand that training large networks, and especially transformers, is resource-intensive and time-consuming. As a middle ground, I suggest a study of pre-trained models (e.g. for ImageNet) that are often trained with dropout. This way, the main claims of the paper can be further backed by an investigation of some large, "real-world" models. It should be easy to reproduce Fig {2,3,4} for pre-trained models as well.
>
> **Response (3)**: We appreciate this suggestion and agree that extending the analysis to pre-trained large-scale models (e.g., ImageNet trained with dropout) would strengthen our framework. We aim to investigate such “real-world” pre-trained networks in future work.
>
> >You should include some experiments with explicit ensembles. So far, you've focused on finding correlations between diversity and performance, but you did not provide an insight about the actual values or value ranges of diversity (i.e. is a diversity of 0.8 good? What does that even mean?). In the classical bias-(co-)variance decomposition, we explicitly know the trade-off between bias, (co-)variance (i.e., diversity), and the overall loss. Your approach currently lacks this. I suggest including experiments with explicit ensembles and comparing diversities here as a first step.
>
> **Response (4)**: We agree that deeper insight into the actual values for each diversity measure would improve the paper. We have added additional discussion to the paper to address this point. With respect to explicit ensembles, we kindly refer the reviewer to Response (2) above.
>
> >You can mention some additional literature, that might be valuable here, especially if you consider explicit ensembles: [1] Generalized Negative Correlation Learning for Deep Ensembling by Buschjäger et al. 2020 (https://arxiv.org/abs/2011.02952) [2] To Ensemble or Not Ensemble: When does End-To-End Training Fail? by Webb et al. 2020 (https://arxiv.org/abs/1902.04422) [3] Learning with Pseudo-Ensembles by Bachman et al. 2014 (https://arxiv.org/abs/1412.4864)
>
> **Response (5)**: Thank you for these references—we appreciate the pointers to related ensemble literature.

---

### Meta-Review · Area_Chair_gevk · 2025-10-31

**Recommendation:** Accept (Poster)
**Confidence:** 4

**Metareview:**

This paper presents an study on the relationship between internal diversity and generalisation, based on the novel premise of viewing a single neural network as an implicit ensemble. While reviewers appreciated the interesting perspective, they raised significant concerns about the limited scope of the empirical evaluation, which used small-scale datasets and models. Additionally, the paper clarity and presentation was raised as a concern that all reviews strongly encouraged the authors to address. However, the authors rebuttal successfully framed the work as a foundational first step and committed to improving the presentation of results, which assuaged most concerns. The consensus, with which the AC agrees, is that despite its preliminary nature, the paper's novel viewpoint is compelling and likely to stimulate further discussion. Therefore, the AC recommends acceptance, strongly encouraging the authors to strengthen the final version by incorporating key results from the appendix into the main text, and improving the clarity where outlined by the reviewers.

---

### Decision · Program_Chairs · 2025-11-05

**Decision:**

Accept (Poster)

**Comment:**

We recommend a poster presentation given the AC and reviewers recommendations.